# Learning to Jointly Share and Prune Weights for Grounding Based Vision and Language Models

**Shangqian Gao**[1*]**, Burak Uzkent**[2]**, Yilin Shen**[2]**, Heng Huang**[1]**, Hongxia Jin**[2]
[1]Department of Electrical and Computer Engineering, University of Pittsburgh
[2]Samsung Research America
{shg84, heng.huang}@pitt.edu,
{b.uzkent, yilin.shen, hongxia.jin}@samsung.com

## Abstract

Transformers have been successful in processing different data modalities, such as language and image data, which could use transformers with similar architectures to achieve good performance. Leveraging this observation, we propose weight sharing across two transformer backbones and within the same transformer backbone and pruning across two backbones in a unified framework. More specifically, we investigate weight sharing and pruning for two components of the transformers: (1) Multi-Head Attention (MSA) and (2) Feed-Forward Network (FFN) layers. To jointly perform weight sharing and pruning, we propose to use a regularization term to align model weights and the desired structure during the multimodal pre-training step. The structure vectors of sharing and pruning are generated by using a hypernetwork, which can capture complex interactions between pruning and sharing across layers and modalities. We train the hypernetwork and model weights iteratively so that the learned structure evolves along with model weights. After minimizing the proposed objective in pre-training step, we perform weight sharing and pruning and fine-tune the compressed model on downstream tasks. Finally, we perform experiments on vision and language tasks, including Referring Expression Comprehension (REC), Visual Question Answering (VQA), and Object Detection using the state-of-the-art grounding based models: MDETR and GLIP. Our experiments show that we can compress these models by $35 - 40\%$ by sharing and pruning MSA and FFN weights without almost any loss in accuracy.

## 1 Introduction

The dominant architecture in natural language processing (NLP) is Transformer (Vaswani et al., 2017). Besides NLP, recent advance in computer vision shows that transformer based model, like ViT (Dosovitskiy et al., 2021) or DeiT (Touvron et al., 2020), can achieve similar or even better performance than convolutional neural networks (CNNs) on various tasks. As a result, it allows us to use architecturally similar models on cross-modal tasks with vision and language data. This setting naturally provides foundations to structurally share weights across different modalities. The advantage of weight sharing is that it encourages weight reuse and thus reduces the number of parameters while maintaining the model capacity to some extent. On the other hand, existing weight sharing techniques have some limitations. Most of them (Lee et al., 2021; You et al., 2022; Lan et al., 2019; Reid et al., 2021) use manually designed sharing rules to share a whole layer or block, largely restricting the flexibility of weight sharing. This reduced flexibility can lead to drastic performance drops. To maximally utilize model parameters, we propose to unify cross-modal sharing, layer-wise sharing, and pruning, all in a single unified framework. Unlike previous works, the minimal structure of these operations is a weight vector instead of a whole layer or block, which drastically increases the flexibility of sharing and pruning. Also, to avoid using manually designed strategies, the positions of sharing and pruning are learned in an end-to-end differentiable manner.

---

[*]Was a research intern in Samsung Research America.

To pursue a better trade-off between the model performance and the parameter efficiency, we aim to maximize flexibility by utilizing the structure of transformer backbones. If only cross-modal sharing is considered, there will be an upper bound for the compression rate ($\sim 50\%$) when sharing all layers of one backbone for another one. Another direction is to share layers within a single backbone similar to Albert (Lan et al., 2019), however, the downside is that there is no prior success of cross-layer sharing for the vision transformer. In addition to weight sharing, pruning is also an option; however, Li et al. (2020); Yu et al. (2022) show that it limits the capacity of the model when the compression rate is high. We argue that, especially in multimodal tasks, it can be hard to achieve a high compression rate while maintaining high accuracy by using only pruning or sharing.

In this direction, we unify cross-modal sharing, layer-wise sharing, and pruning into a single framework to maximize the flexibility for sharing and pruning. With the increased flexibility, finding which weights to prune or share can be very difficult. To solve this problem, we design a hypernetwork for learning the binary structure vectors for sharing and pruning efficiently. The hypernetwork can effectively capture the complex interactions between pruning and sharing across different layers and modalities. Since we can not share and prune a weight vector at the same time, we apply constraints on different parts of structure vectors. We use a regularization term to softly align the learned structures and backbone weights instead of directly sharing and pruning by applying structure vectors. The benefit of such a regularization term is that we do not need to add a separate fine-tuning process to the whole pre-training step. Finally, we iteratively train the hypernetwork and the model so that the learned structures can adapt to the changes in the model during training. With these novel designs, our method learns suitable structures for sharing and pruning with small extra costs.

We evaluate the effectiveness of our method with two state-of-the-art vision and language grounding models: MDETR (Kamath et al., 2021), GLIP (Li et al., 2022). We use the CLIP text transformer (Radford et al., 2021) and DeiT (Touvron et al., 2020) as the text backbone and the vision backbone, and thus two backbones have similar architectures. Our method is evaluated on different downstream tasks: RefCOCO/RefCOCO+ (Yu et al., 2016)/RefCOCOg (Mao et al., 2016) for referring expression comprehension (REC), GQA (Hudson & Manning, 2019) for visual question answering (VQA), MS-COCO object detection (Lin et al., 2014) and Flickr-30k (Plummer et al., 2015) for phrase grounding. On these benchmarks, our method can remove around $40\% \sim 45\%$ parameters of the backbones and $35\% \sim 40\%$ of all the parameters with almost no accuracy drop. In some tasks, our method even outperforms the original model. These results show that the proposed framework achieves a prominent trade-off between the number of parameters and accuracy.

## 2 RELATED WORKS

**Vision and Language Models.** Existing vision and language models can be broadly grouped into two categories: (1) two-stage and (2) single-stage. Two-stage methods (Yu et al., 2018; Chen et al., 2020; Lu et al., 2019) utilize off-the-shelf object detectors, e.g., Faster-RCNN, to detect objects and represent them with convolutional features. For referring expression comprehension, the referring expression is matched to its appropriate region (Yu et al., 2018). For other vision and language tasks, region descriptions are passed through two transformers to get image and language representations (Lu et al., 2019; Zhang et al., 2021a; Gan et al., 2020). Single-stage methods (Kamath et al., 2021; Deng et al., 2021; Yang et al., 2020; Chen et al., 2018; Li & Sigal, 2021) avoid using a detached off-the-shelf object detector and perform end-to-end training and inference, reducing the computational complexity of the two-stage methods. Another advantage of single-stage methods is that the full model can be pre-trained for text-conditioned object detection as in MDETR (Kamath et al., 2021), GLIP (Li et al., 2022) GLIP-v2 (Zhang et al., 2022), etc.

**Pruning with Transformers.** With the increasing popularity of vision transformers, there are many works that perform pruning on different components, such as pruning tokens (Kong et al., 2021; Xu et al., 2021; Rao et al., 2021), pruning structures or weights (Chen et al., 2021; Yin et al., 2023; Lou et al., 2022b), or all of the above (Yang et al., 2021). The pruning for language transformers (Li et al., 2020; Gale et al., 2019) can be traced back even earlier. Unlike these works, in the cross-modal setting, we only prune weights that are not useful in both modalities.

**Transformer-based Unified Backbones.** Transformers recently enjoyed tremendous popularity in processing different modalities, including images, videos (Girdhar et al., 2019), speech (Dong et al., 2018), audio (Jaegle et al., 2021), and language (Jaegle et al., 2021; Hu & Singh, 2021; Zhang et al., 2021b). The universality of transformers has been exploited for multi-tasking using the same vision backbone for different vision-only tasks (Girdhar et al., 2022), the same vision and language

backbone for different vision and language tasks (Hu & Singh, 2021). Likhosherstov et al. (2021), on the other hand, performs multitasking with the same transformer backbone on different modalities, and Wang et al. (2021) pre-trains the model on vision-only, text-only, and vision-language tasks and fine-tune the model on vision or text-only or vision-language tasks. Similarly, Li et al. (2021); You et al. (2021) pre-train a unified transformer backbone with images and their captions and fine-tune the backbone for the single modality downstream tasks. Different from these studies, we propose *structural* cross-modal, layer-wise sharing and pruning across two transformer backbones for grounding based vision and language downstream tasks with almost no loss in accuracy.

**Weight Sharing with Transformers.** There has been a number of studies that focus on sharing weights in a single transformer on language-only tasks (Lan et al., 2019; Reid et al., 2021; Takase & Kiyono, 2021; Lou et al., 2022a). These studies mostly re-use a language transformer's full encoder (MSA+FFN) at different depths using some manually defined strategy. Parameter efficient cross-modal transformers (Lee et al., 2021) reduces parameters with manually designed weight sharing strategies. Another recent study, Kim et al. (2022), performs weight sharing in the multimodal fusion network, which can be orthogonal to our study as we explore weight sharing across the transformer backbones. Finally, a recent work, MS-CLIP (You et al., 2022), shares weights across modalities by rigorously examining the architecture choices and adding early specialization layers. On the other hand, our method is general and flexible as we learn where to share and prune weight vectors in a vision and language model. Additionally, unlike MS-CLIP, we do not change the architecture of the original model. As a result, our method can be applied to any vision and language model with transformer backbones. Finally, different from these studies, we focus on grounding based vision and language models, which have shown impressive performance on many vision and language tasks (Kamath et al., 2021; Li et al., 2022; Zhang et al., 2022).

## 3 LEARNING TO SHARE AND PRUNE STRUCTURES FOR VISION AND LANGUAGE TRANSFORMERS

### 3.1 OVERVIEW

In a typical vision and language model, we have two backbones: a vision backbone ($v$) and a text backbone ($t$). We use $W^{*_l} * \in \{v, t\}$ to denote the weight matrix of $l$'th layer from the vision ($W^{v_l}$) or the text backbone ($W^{t_l}$), and $l = 1, \ldots, L$. $L$ is the total number of layers. This definition also applies to sharing and pruning vectors: $s^*$ and $m^*$. Since we consider architecturally similar backbones, the vision and text backbone have the same $L$.

Usually, transformers are used for processing text and vision data. A classic transformer has two core components: (1) Multi-Head Attention (MSA) and (2) Feed-Forward Network (FFN) layers. They are the targets for structural weight sharing and pruning since these components contain most of the weights in backbones. To find which structures to prune or share, we first need to optimize the sharing and pruning vectors produced by the hypernetwork during the pre-training process of MDETR (Kamath et al., 2021) and GLIP (Li et al., 2022). To further increase the flexibility of the pruning and sharing process, layer-wise sharing (inspired by Albert (Lan et al., 2019) but at a fine-grained level) across the text backbone is also performed. A detailed comparison of different sharing and pruning options is given in Fig. 2. Since we learn discrete structure vectors, directly applying them in the fine-tuning step can result in severe accuracy drops. We instead softly align model weights and learned structures via a regularization term in the pre-training step. After the pre-training step, we conduct structural pruning and sharing based on learned structure vectors. Finally, we fine-tune the compressed model for the downstream task. An overview of how our method shares and prunes weights is provided in Fig. 1.

### 3.2 SHARING AND PRUNING ACROSS TWO BACKBONES

For MSA layers, we have three weight matrices for query $W_q^* \in R^{d \times d}$, key $W_k^* \in R^{d \times d}$ and value $W_v^* \in R^{d \times d}$ and again $* \in \{v, t\}$. With input tokens $X^* \in R^{N^* \times d}$, the final $Q^*$, $K^*$ and $V^*$ for self-attention is obtained by:

$$Q^* = X^* W_q^*, \ K^* = X^* W_k^*, \ V^* = X^* W_v^*, \tag{1}$$

where $d$ is the embedding dimension, $N^*$ represents the number of tokens given the modality. To make minimal changes to the original model, we can perform pruning on $W_q^*$ and $W_k^*$, and structural sharing are applied to all three weight matrix between backbones and across layers.

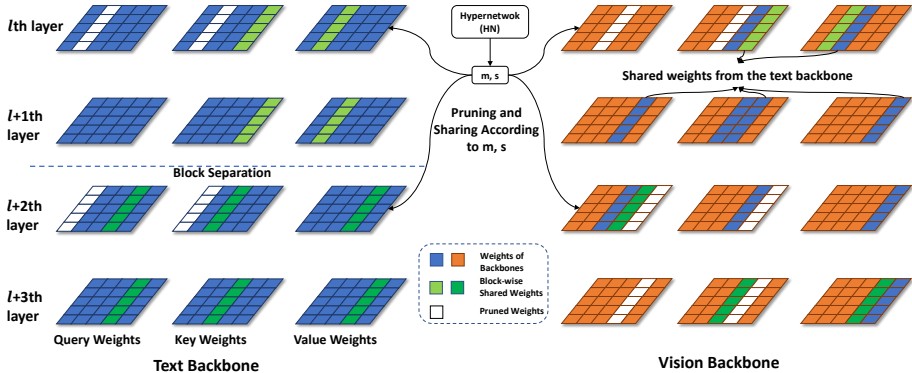

Figure 1: An overview of weights sharing and pruning across two backbones. Here, for simplicity, we use MSA layers as an example, and the process for the FFN layer is similar. The hypernetwork (HN) generates $m, s$ to guide the pruning and sharing across modalities and within layers in a block. As shown in the figure, to incorporate block-wise sharing, the text backbone is separated into several blocks. The conflicts between sharing and pruning are resolved before they are applied to backbones' weights. Note that different from other weights, $W_v$ is not pruned.

For pruning, we use a binary vector $m^* \in \{0, 1\}^d$ on $W_q^*$ and $W_k^*$. Let's use the vision backbone as an example for pruning. The production of the query and key when calculating MSA then becomes:

$$Q^{v_l}(K^{v_l})^T = X^{v_l}(W_q^{v_l} \odot m^{v_l})(W_k^{v_l} \odot m^{v_l})^T(X^{v_l})^T, \tag{2}$$

where $m^{v_l}$ is first resized to have the same dimension of $W_q^{v_l}$ and $\odot$ is the element-wise production. $X^{v_l}$ is the feature map from the previous layer.

We perform cross-modal sharing between two backbones, and cross-layer sharing for the text backbone. To achieve sharing, we use a binary vector $s^* \in \{0, 1\}^d$ to share weights between two modalities and different layers. Weights of the text backbone are used for cross-modal sharing. As a result, a weight vector could be implicitly used in different layers across text and vision backbones. Let's use $W_q^{v_l}$ as an example for cross-modal sharing:

$$W_q^{v_l} = s^l \odot W_q^{v_l} + (1 - s^l) \odot W_q^{t_l}, \tag{3}$$

where $s^l$ is also first expanded to have the same size of $W_q^{v_l}$. In short, weight vectors of index $i$ with $s_i^l = 0$ are shared. Without loss of generality, we omitted the modal notation ($v$ or $t$) for cross-modal sharing $s^l$, since the sharing direction is fixed. Similarly, cross-layer sharing has the following setup:

$$W_q^{t_l} = s^{t_l} \odot W_q^{t_l} + (1 - s^{t_l}) \odot W_q^{t_b}, \tag{4}$$

where $W_q^{t_l}$ represents the weights of $l$th layer in the text backbone, $W_q^{t_b}$ is the assigned base weights, and $s^{t_l}$ is the vector used for cross-layer sharing. As a result, the final weights for the text backbone is divided into two parts: layer-specific weights and shared weights from base layers.

A naive way for cross-layer sharing uses weights from a certain layer as base weights for all other layers (Lan et al., 2019). However, it can restrict the diversity of weights across layers. To enhance the flexibility of cross-layer sharing, we divide the text backbone into several blocks, and each block has its own base weights. Specifically, we establish a set of base layers (in sorted order) $B = \{b_1, b_2, \cdots, b_{|B|}\}$. All layers are naturally split into several blocks based on base layers, e.g., the first block contains layers with $b_1 \leq l < b_2$. In our experiments, the base weight is set to the first layer of each block. When $l$ is a base layer ($l \in B$), we force $s^{t_l} = 1$ since we assign the base layer its own weights and potentially reuse it for the other layers in the same block.

Note that after cross-layer sharing, the base weights for cross-modal sharing is also changed. By putting cross-layer sharing and cross-modal sharing together, we have:

$$W_q^{v_l} = s^l \odot W_q^{v_l} + (1 - s^l)s^{t_l} \odot W_q^{t_l} + (1 - s^l)(1 - s^{t_l}) \odot W_q^{t_b}. \tag{5}$$

As a result, the final $W_q^{v_l}$ consists weights from $W_q^{t_l}$ (weights from the text backbone of the same layer), $W_q^{t_b}$ (weights from the text backbone of the base layer) and vision modality specific weights.

Figure 2: Overview of different settings of sharing and pruning. By including cross-modal, block-wise sharing, and pruning, we can reduce the number of parameters in different ways.

For FFN layers, we have a weight matrix $W_E^* \in R^{d \times d'}$ expanding the feature dimension, and another weight matrix $W_C^* \in R^{d' \times d}$ compressing the feature dimension. To keep the feature map size $d$ unchanged, we conduct sharing and pruning along the dimension with size $d'$ and the process is similar to MSA layers. The detailed formulation is provided in the supplementary materials. We do not apply sharing and pruning for other layers including Layer Norm (Ba et al., 2016) since they contain a very small amount of parameters.

### 3.3 CONFLICTS OF STRUCTURE VECTORS

Without any constraint, $m$ and $s$ could have conflicts. For example, if we prune weights, it becomes meaningless to share the same weights. To resolve such conflicts, we impose the following constraint for cross-modal sharing and pruning:

$$(m_i^{v_l}, s_i^l) \in C, \ (m_i^{t_l}, s_i^l) \in C, \text{ where } C = \{(x, y)|(x, y) \neq (0, 0)\}, \tag{6}$$

where $i$ is the index of sharing or pruning for a certain weight vector. If $(m_i, s_i) \in C$, then the structure vector does not share and prune the same weight vector which resolves the conflicts. This constraint can also be applied for cross-layer sharing and pruning for the text backbone, but in a more sophisticated way:

$$\begin{cases} (m_i^{t_l}, s_i^{t_l}) \in C, \text{ where } C = \{(x, y)|(x, y) \neq (0, 0)\}, \ l \notin B, \\ (m_i^{t_l}, s_i^{t_l} s_i^{t_{l+1}} \cdots s_i^{t_{b'}}) \in C, \text{ where } C = \{(x, y)|(x, y) \neq (0, 0)\}, \ l \in B, \end{cases} \tag{7}$$

where $b'$ represents the last element of the current block. For example, if $l = b_1$, the current block consists layers of $b_1 \leq l < b_2$, and $b' = b_2 - 1$. With $s_i^{t_l} s_i^{t_{l+1}} \cdots s_i^{t_{b'}}$ we represent all the shared elements from different layers in the current block, and these elements should be kept in the base layer.

We do not add this constraint between $s_i^l$ and $s_i^{t_l}$, since there is no conflict between sharing. To directly apply this constraint, we can prioritize $m_i$ or $s_i$ and set the other one manually. For example, let $m_i^{t_l} = s_i^l = 0$, we can directly set $m_i^{t_l} = 1$ to comply with the constraint in Eq. 6 if $s_i$ is prioritized (shared weights will not be pruned). In practice, we prefer sharing weights to pruning weights, since sharing preserves the model capacity to some extent.

### 3.4 LEARNING STRUCTURE VECTORS

To generate $m, s$, we use a hypernetwork (HN) parameterized by $\theta$ and Gumbel-Sigmoid (Jang et al., 2016) technique:

$$m, s = \text{HN}(z, \theta), \tag{8}$$

where $z$ is the input to the HN, and it is a predefined vector. Basically, the HN is composed of GRUs (Chung et al., 2014) and multilayer perceptrons (MLPs). GRU can capture inter-layer interactions, and MLP is used for intra-layer interactions. More details of the HN are presented in the supplementary materials. To learn the HN, we want to solve the following optimization problem:

$$\min_{\theta} \ \mathcal{L}_{\text{pre-training}}(x, y; m, s) + \lambda \mathcal{R}(P(m, s), p P_{\text{total}}), \tag{9}$$

where $x$ is the input sample of the image and text pair, $y$ is its label, $\mathcal{L}_{\text{pre-training}}$ is the original pre-training loss from MDETR (Kamath et al., 2021) or GLIP (Li et al., 2022), and its model structure is

---

**Algorithm 1:** Learning to Jointly Share and Prune Weights for Grounding Based Vision and Language Tasks

---

**Input**: the pre-training dataset and a sub-dataset for learning structure vectors: $D, D_{\text{sub}}$;
  remained rate of parameters: $p$; hyper-parameter: $\lambda, \gamma$; pre-training epochs: $E$; the model for
  pre-training: $f$; the hypernetwork HN parameterized by $\theta$

**for** $e := 1$ *to* $E$ **do**

  | /\* Optimizing model weights. Freeze $\theta$ in HN             \*/
  | **for** *a mini-batch* $(x, y)$ *in* $D$ **do**
  |   | 1. generate $m, s$ from HN with Eq. 8.
  |   | 2. apply constraints on $m, s$ defined in Eq. 6 and Eq. 7, and sharing is prioritized.
  |   | 3. calculate $\mathcal{R}_w(W, m, s)$ given $W$ and $m, s$.
  |   | 4. calculate gradients *w.r.t* $W$ by minimizing Obj. 11 and update $W$.
  | **end**
  | /\* Optimizing HN weights. Freeze $W$ in the model.         \*/
  | **for** *a mini-batch* $(x, y)$ *in* $D_{sub}$ **do**
  |   | 1. generate $m, s$ from HN with Eq. 8.
  |   | 2. apply constraints on $m, s$ defined in Eq. 6 and Eq. 7, and sharing is prioritized.
  |   | 3. calculate the parameter regularization term $\mathcal{R}(P(m, s), pP_{\text{total}})$.
  |   | 4. calculate gradients *w.r.t* $\theta$ by minimizing Obj. 9 and update $\theta$.
  | **end**

**end**
Get $f'$ by pruning and sharing $f$ based on $m, s$. **Return** $f'$ for task-specific fine-tuning.

---

decided by $m, s$. On the other hand, $\lambda$ controls the strength of $\mathcal{R}(P(m, s), pP_{\text{total}})$. The regularization loss, $\mathcal{R}(P(m, s), pP_{\text{total}})$, controls how much parameters we should keep given by $p \in (0, 1]$. $P(m, s)$ in the regularization loss represents the remaining number of parameters decided by $m, s$ and $P_{\text{total}}$ is the total number of parameters of MSA+FFN layers in backbones. In practice, we let $\mathcal{R}(P(m, s), pP_{\text{total}}) = \log(\max(P(m, s), pP_{\text{total}})/pP_{\text{total}})$ (Gao et al., 2021). On the other hand, $\mathcal{R}$ could be any regression loss functions, including MSE and MAE; however, it might be harder to push $\mathcal{R}$ small enough, especially when the number of parameters is large.

During the optimization of the HN, we keep the weights of the whole model including backbones and the fusion part frozen. Since the time cost for pre-training is quite large, we use a small portion of the whole dataset to train HN. Next, we train model weights on the whole pre-training dataset while controlling its architecture with $m, s$ by training the hypernetwork on a subset iteratively.

## 3.5   Learning to Jointly Share and Prune Weights

To share and prune the weights, we can directly apply $m$ and $s$ (from the hypernetwork) to vision and text transformers. However, it can drop the accuracy since the outputs of the vision and text backbones are drastically changed. Thus, to alleviate this problem, we use a selection based regularization mechanism to softly push selected weights closer (for sharing) or to zeros (for pruning):

$$\mathcal{R}_w(W, m, s) = \sum \|(1 - s) \odot W^t - (1 - s) \odot W^v\|_1$$
$$+ \sum \|(1 - s^{t_l}) \odot W^{t_l} - (1 - s^{t_l}) \odot W^{t_b}\|_1 + \sum m_i \cdot \|W_{[:,i]}\|_2, \tag{10}$$

where the first two terms push selected weight vectors closer and the final term uses Group Lasso to push selected weight vectors to $0$. With this regularization term, it gradually aligns model weights to the learned structure vectors which creates a smooth process for reducing the number of model parameters. Given the regularization loss, we learn model weights $W$ by optimizing the following objective function:

$$\min_W \mathcal{L}_{\text{pre-training}}(x, y; W) + \gamma \mathcal{R}_w(W, m, s), \tag{11}$$

where $\gamma$ controls the strength of $\mathcal{R}_w(W, m, s)$. After the pre-training process, the corresponding weights are pruned and shared to compress the model. The compressed model can then be used for fine-tuning on downstream tasks.

We present the training process of the proposed method in Algorithm. 1. Note that (1) the forward calculation is different when optimizing the model (Obj. 11) and HN (Obj. 9). When optimizing HN, $m, s$ are applied in the forward calculation, and when optimizing the model, the forward calculation

| Settings | RefCOCO | RefCOCO+ | RefCOCOg | GQA | #PT | #PB |
|---|---|---|---|---|---|---|
| DeiT-Tiny+CLIP | 75.1 | 65.3 | 67.7 | 50.9 | 53M | 44M |
| DeiT-Small+Albert | 75.3 | 65.2 | 67.7 | 51.0 | 48M | 34M |
| Backbones: DeiT-Small+CLIP | | | | | | |
| Baseline | 77.5 | 67.3 | 69.9 | 52.9 | 72M | 63M |
| FS | 70.1 | 59.6 | 61.5 | 44.7 | 50M | 41M |
| SP | 76.2 | 65.8 | 68.5 | 52.3 | 52M | 43M |
| LWSP | 77.2 | 66.5 | 68.6 | 52.8 | 48M | 39M |
| BWSP | 77.6 | 67.3 | 69.4 | 53.0 | 48M | 39M |
| BWSP w/o $R_w$ | 71.6 | 61.4 | 62.2 | 46.4 | 48M | 39M |

Table 1: Results on tasks from MDETR. We use MDETR-Small with CLIP+DeiT-Small backbones. #PT/#PB in this table, Tab. 2, Tab. 3 and Tab. 4 represent the number of total parameters and backbone parameters.

| Settings | RefCOCO | RefCOCO+ | RefCOCOg | GQA | #PT | #PB |
|---|---|---|---|---|---|---|
| DeiT-Tiny+CLIP | 76.8 | 69.9 | 70.0 | 54.3 | 141M | 124M |
| DeiT-Small+CLIP | 77.5 | 70.6 | 70.3 | 54.7 | 159M | 142M |
| Backbones: DeiT-Base+CLIP | | | | | | |
| Baseline | 79.8 | 72.1 | 72.9 | 56.7 | 226M | 209M |
| FS | 72.8 | 62.6 | 63.0 | 46.1 | 141M | 124M |
| BWSP | 80.5 | 72.4 | 72.6 | 56.6 | 136M | 119M |
| BWSP w/o $R_w$ | 73.9 | 63.4 | 64.6 | 48.1 | 136M | 119M |

Table 2: Results on tasks from MDETR. We use MDETR-Large with CLIP+DeiT-Base backbones.

remains intact. (2) To reduce the training time of the HN, we use a small subset $D_{sub}$ of the pre-training dataset, $D_{sub} \ll D$. (3) The goal of using HN is to accelerate the learning of $m, s$, and better capture the complicated interaction between pruning and sharing across modalities and layers. We note that we can also directly set $m, s$ as learnable parameters, however, it can slow down the learning process and thus lower the final performance.

## 4 EXPERIMENTS

### 4.1 SETTINGS

**GLIP and MDETR.** We apply our method to two recently published grounding based vision and language models: (1) MDETR (Kamath et al., 2021) and (2) GLIP (Li et al., 2022). MDETR performs end-to-end pre-training on dataset of 1.3M image-text pairs for the task of text-conditioned object detection. On the other hand, GLIP scales up the pre-training dataset for text-conditioned object detection to 27M image-text pairs by training a teacher network on 1.3M image-text pairs with ground truth bounding boxes and using the teacher on 27M image-text pairs to get pseudo bounding boxes. Pre-trained GLIP and MDETR are then used for vision and language tasks.

**Baselines.** We constructed several alternative settings to verify the design choice of our method. The first setting is **F**ully cross-modal **S**haring (**FS**). For FS, we share all MSA+FFN layers across two backbones during both the pre-training stage and the fine-tuning stage without any regularization term. We note that FS can be considered as a manually designed weight sharing method similar to You et al. (2022); Lee et al. (2021). The second setting is cross-modal **S**haring and **P**runing (**SP**). For SP, we only apply cross-modal sharing and pruning for the vision and text backbones. The third setting is cross-modal, **L**ayer-**W**ise **S**haring and **P**runing (**LWSP**). On top of SP, LWSP also shares weights across layers in the text backbone. The third setting is cross-modal, **B**lock-**W**ise **S**haring and **P**runing (**BWSP**). For BWSP, the text backbone is split into three blocks, and each block contains 4 consecutive MSA and MLP layers. Fig. 2 shows the concept of these settings. By default, layer-wise sharing and block-wise sharing are applied for the text backbone, and the weights of the text backbone are reused for the vision backbone for the cross-modal sharing since Lu et al. (2021) showed that language transformers can generalize to vision tasks.

**Implementation Details.** In all experiments, we replace the original backbones used in MDETR and GLIP with the CLIP text transformer and DeiT to have architecturally similar backbones. Due to the resource limitation, we change the GLIP pre-training dataset to be the same as MDETR, and the pre-training dataset for MDETR is not changed. We note that MDETR and GLIP train the models on different size images and test the models on images with smaller size resized to 800 pixels. On the other hand, we resize the images to 224x224 pixels in training and test time as the motivation of

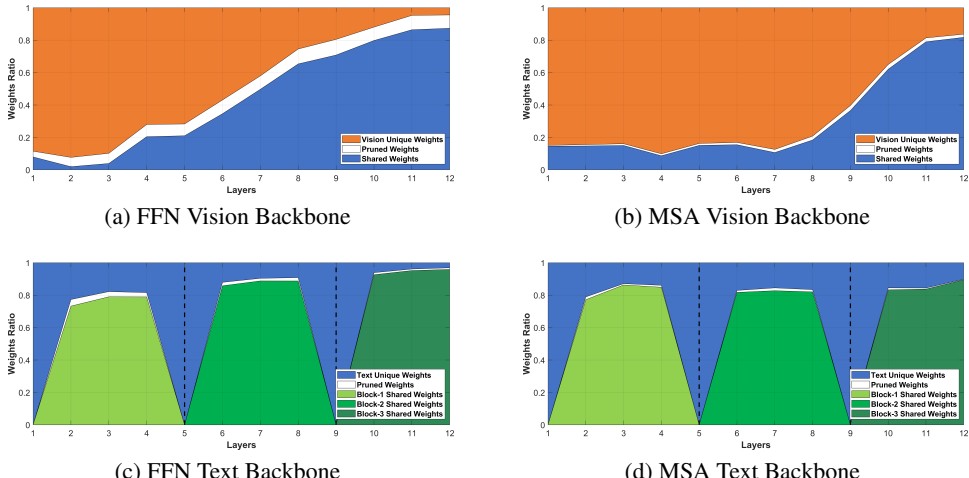

Figure 3: Ratios of weight sharing and pruning across the MSA and FFN layers in two backbones. Dashed lines in the text backbone represent block separations.

| Settings | AP | $AP_{50}$ | $AP_{75}$ | #PT | #PB |
|---|---|---|---|---|---|
| Backbones: CLIP+DeiT-Small | | | | | |
| Baseline | 20.0 | 32.7 | 20.7 | 83M | 63M |
| SP | 19.1 | 31.0 | 19.7 | 63M | 43M |
| LWSP | 19.4 | 31.4 | 20.1 | 60M | 40M |
| BWSP | 19.9 | 32.4 | 20.4 | 57M | 37M |

Table 3: COCO object detection results. We use GLIP with CLIP+DeiT-Small as backbone.

our study is to maintain the accuracy of the full model after compression. Due to space limitation, other implementation details are given in supplementary materials.

## 4.2 RESULTS

**MDETR results.** For MDETR, we construct two models, MDETR-Small and MDETR-Large, with different model settings. We report results on the validation dataset of Ref-COCO/RefCOCO+/RefOCOCg and GQA. The result of MDETR-Small is shown in Tab. 1. From the table, we can see that the performance of BWSP is on par with the baseline model (relative changes: +0.1/+0.0/-0.5/+0.1 for four tasks). At the same time, BWSP reduces 24M parameters (around 38% of two backbones) compared to the baseline model. LWSP and SP perform worse than BWSP, indicating that increasing flexibility is helpful when reducing the number of parameters. The results of MDETR-Large is shown in Tab. 2. On MDETR-Large, BWSP also performs similarly to the baseline model, but the parameter reduction rate is larger (40% of the baseline model). For both MDETR-Large and MDETR-Small, the manually designed sharing rule FS causes significant performance drops. In addition, given a similar number of parameters, BWSP outperforms other full models with different backbones on both the large and small model setups. Finally, we remove $R_w$ during pre-training, and we directly train the hypernetwork and fine-tune the model. We call this setting BWSP w/o $R_w$. As expected, BWSP w/o $R_w$ results in a much larger performance drop, which justifies the design choice of using a $R_w$ to align model weights and structures during pre-training.

**GLIP results.** We evaluate our method with GLIP on MS-COCO object detection (COCO) and Flickr-30k for phrase grounding. The results on COCO are shown in Tab. 3. From the table, we can see that BWSP still outperforms SP and LWSP. Besides the performance, BWSP also removes more parameters. In Tab. 4, we can have a similar observation. In summary, BWSP provides more flexibility than SP and LWSP, which results in a stronger performance/parameter trade-off.

## 4.3 DETAILED ANALYSIS

**Effects of $R_w$.** In Fig. 4, we plot the weight difference with or without $R_w$. Let's consider cross-modal sharing as an example, for $l$th layer, we plot $\frac{1}{n}\sum_i^n \|(1-s_i) \odot W^{t_l} - (1-s_i) \odot W^{v_l}\|_1$ for regularized weights, and we plot $\frac{1}{n}\sum_i^n \|s_i \odot W^{t_l} - s_i \odot W^{v_l}\|_1$ for weights without regularization. The figure shows that the weight difference is much smaller with regularization for both pruning and cross-modal sharing. For layer-wise sharing, the difference is smaller since the ratio of layer-shared

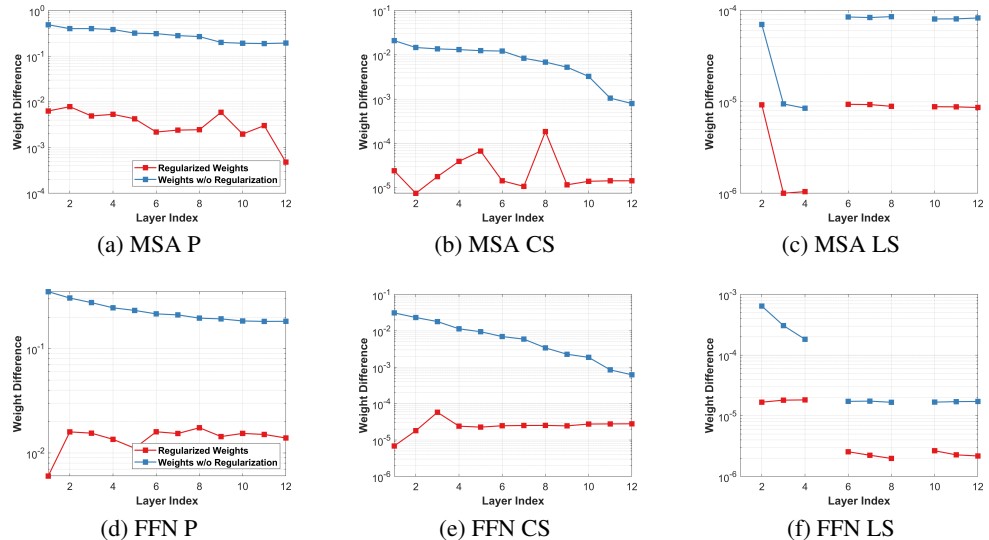

Figure 4: Weights difference with or without the regularization $R_w$. P represents **P**runing, CS represents **C**ross-modal **S**haring, and LS represents **L**ayer-wise **S**haring.

| Settings | Val | | | Test | | | #PT | #PB |
|---|---|---|---|---|---|---|---|---|
| | R@1 | R@5 | R@10 | R@1 | R@5 | R@10 | | |
| Backbones: CLIP+DeiT-Small | | | | | | | | |
| Baseline | 72.4 | 86.9 | 89.6 | 72.5 | 86.9 | 89.5 | 83M | 63M |
| SP | 71.6 | 86.4 | 89.0 | 71.8 | 86.3 | 89.3 | 63M | 43M |
| LWSP | 72.2 | 86.9 | 89.4 | 72.3 | 86.6 | 89.5 | 60M | 40M |
| BWSP | 72.4 | 86.9 | 89.4 | 72.7 | 86.7 | 89.5 | 57M | 37M |

Table 4: Flickr30k results. We use GLIP with CLIP+DeiT-Small backbones.

weights is larger. In conclusion, Fig. 4 shows that the proposed regularization $R_w$ can effectively encourage sharing and pruning of desired weights, and learning other weights is almost intact.

**Analysis of Pruned and Shared Weights.** In Fig. 3, we plot the details of pruned and shared weights of BWSP for MDETR-Small. We observe that the layer-wise sharing rate is large for both FFN and MSA layers of the text backbone. For the vision backbone, earlier layers are almost preserved for both FFN and MSA layers. We also observe that FFN layers from the vision backbone are shared more aggressively whereas MSA layers are only shared for later layers. On the other hand, the ratio of pruning is not obvious except for the vision FFN layers. In summary, the text backbone contributes more to the reduction of weights, which is reasonable since vision tasks are more complex than language tasks in general for the tasks required for MDETR and GLIP models.

## 5 CONCLUSION

In this paper, we investigate how to perform structural sharing and pruning for grounding based vision and language models. Specifically, we perform cross-modal sharing and pruning across the vision and text backbones and layer-wise sharing in the text backbone. With larger flexibility, it becomes harder to select the right set of weight vectors to share and prune. To overcome this challenge, we propose an hypernetwork to capture the interactions of pruning and sharing across layers and modalities. Next, in the pre-training stage, we gradually align the backbones' weights with the desired architecture learned by the hypernetwork. Finally, the model is fine-tuned for downstream tasks. Our experiments on MDETR and GLIP show that we can reduce $35\% \sim 40\%$ of model weights without a significant loss in accuracy.

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

# A SUPPLEMENTARY MATERIALS

## A.1 PRUNING AND SHARING FOR FFN LAYERS

We will provide more details for the pruning and sharing of FFN Layers in this section. Recall that we have a weight matrix $W_E^* \in R^{d \times d'}$ expanding the feature dimension, and another weight matrix $W_C^* \in R^{d' \times d}$ compressing the feature dimension for FFN layers. To keep the model dimension $d$ unchanged, we conduct sharing and pruning along the dimension with size $d'$. For the cross-modal sharing, we then have:

$$
\begin{aligned}
W_E^{v_l} &= s^l \cdot W_E^{v_l} + (1 - s^l) \cdot W_E^{v_l}, \\
W_C^{v_l \, T} &= s^l \cdot W_C^{v_l \, T} + (1 - s^l) \cdot W_C^{v_l \, T},
\end{aligned}
\tag{12}
$$

like MSA layers, $s^l$ is first resized to have the same size of $W_E^{v_l}$. For layer-wise sharing, we have the following equations:

$$
\begin{aligned}
W_E^{t_l} &= s^{t_l} \odot W_E^{t_l} + (1 - s^{t_l}) \odot W_E^{t_b}, \\
W_C^{t_l \, T} &= s^{t_l} \odot W_C^{t_l \, T} + (1 - s^{t_l}) \odot W_C^{t_b \, T}.
\end{aligned}
\tag{13}
$$

$s^{t_l}$ is resized to have the same size of $W_E^{t_l}$. Let's take the vision backbone as an example, the pruning for $W_E^{v_l}$ and $W_C^{v_l}$ is done by inserting the mask $m^{v_l}$ to the intermediate feature map between $W_E^{v_l}$ and $W_C^{v_l}$: $\hat{X}^{v_l} = m^{v_l} \odot X^{v_l}$. To simplify notations, we reuse the notations $s^l$, $s^{t_l}$ and $m^{v_l}$ for FFN layers. Similar to MSA layers, sharing and pruning for FFN layers also satisfy the constraints defined in Eq. 7 and Eq. 6.

## A.2 DETAILED STRUCTURE OF HYPERNETWORK

The detailed architecture is summarized in Tab. S1. The input of the hypernetwork $z \in R^{2L \times 32}$,

Table S1: The architecture of the hypernetwork.

| Input $z$ |
| --- |
| GRU(32,64)$\rightarrow$ LayerNorm$\rightarrow$ GeLU |
| MLP$_l$(64, $d_s^l$)$\rightarrow$ Outputs $o_s^l$, $l = 1, \cdots, L$
MLP$_l$(64, $d_m^{v_l}$)$\rightarrow$ Outputs $o_m^{v_l}$, $l = 1, \cdots, L$
MLP$_l$(64, $d_s^{t_l}$)$\rightarrow$Outputs $o_s^{t_l}$, $l = 1, \cdots, L$
MLP$_l$(64, $d_m^{t_l}$)$\rightarrow$ Outputs $o_m^{t_l}$, $l = 1, \cdots, L$ |

and is sampled from a uniform distribution and it is kept unchanged after the training starts. The outputs of GRUs are parallelly fed into MLP layers to get the outputs before the final pruning and sharing vectors. The final $s^l$, $m^{v_l}$, $s^{t_l}$, $m^{t_l}$ is obtained by using the Gumbel-Sigmoid (Jang et al., 2016) technique:

$$
\begin{aligned}
s^l &= \text{round}(\text{sigmoid}((o_s^l + g + b)/\tau)), \\
m^{v_l} &= \text{round}(\text{sigmoid}((o_m^{v_l} + g + b)/\tau)), \\
s^{t_l} &= \text{round}(\text{sigmoid}((o_s^{t_l} + g + b)/\tau)), \\
m^{t_l} &= \text{round}(\text{sigmoid}((o_m^{t_l} + g + b)/\tau)),
\end{aligned}
\tag{14}
$$

where sigmoid$(\cdot)$ is the sigmoid function, round$(\cdot)$ is the function that rounds its input to its nearest integer, $g$ follows Gumbel distribution: $g \sim \text{Gumbel}(0, 1)$, $b$ is a constant and $\tau$ is the temperature hyper-parameter. We set $b = 3.0$ so that model structures are kept intact at the beginning. The $\tau$ is set to $0.4$ for all experiments. Since round$(\cdot)$ is not differentiable, we use the straight-through gradient estimator (Bengio et al., 2013) to circumvent this problem.

$L$ is the total number of MSA and FFN layers, and the order of MSA and FFN layers is followed by their natural order in the transformer. Recall that the model embedding dimension is $d$, and $d'$ is the dimension after expansion for FFN layers. The dimension of each MLP is decided by whether it is MSA or FFN layers. For sharing operations, $d_s^l$ and $d_s^{t_l}$ equal to $3d$ when they are MSA layers. For pruning operations, $d_m^{v_l}$ and $d_m^{t_l}$ are equal to $d$, and they are shared for query weights and key weights. For FFN layers, $d_s^l$, $d_m^{v_l}$, $d_s^{t_l}$ and $d_m^{t_l}$ equal to $d'$.

## A.3 ADDITIONAL DETAILS FOR EXPERIMENTS

| Model | Backbones | Settings | $p$ |
|-------|-----------|----------|-----|
| MDETR | CLIP+DeiT-Small | SP | 0.5 |
|       |                 | LWSP | 0.4 |
|       |                 | BWSP | 0.4 |
|       | CLIP+DeiT-Base | BWSP | 0.4 |
| GLIP | CLIP+DeiT-Small | SP | 0.5 |
|      |                 | LWSP | 0.4 |
|      |                 | BWSP | 0.33 |

Table S2: Choice of $p$ for different models and settings.

The hyper-parameters $\lambda$ and $\gamma$ for $\mathcal{R}$ and $\mathcal{R}_w$ are set to $6.0$ and $0.0005$ for all datasets and tasks. We train the hypernetwork with ADAM and a constant learning rate of $0.001$ and weight decay $0.0001$. For $D_{\text{sub}}$, we random sample $10\%$ of samples from $D$. We found that this hyper-parameter setting performs well for both MDETR and GLIP. Other pre-training and fine-tuning settings are identical to the original setting. In our method, $p$ controls how much compressible backbone parameters should be removed. We list the detailed choice of $p$ in Tab. S2. We only need to modify $p$ if we want to change how much weight we want to compress. The results of all MDETR tasks are obtained by fine-tuning on downstream tasks. Results of GLIP on COCO detection task Tab. 3 are also obtained by fine-tuning. We follow the fine-tuning setting in their original papers and codes[1]. Let's take GLIP pre-taining as an example. We use the hyper-parameters given in https://github.com/microsoft/GLIP/blob/main/configs/pretrain/glip_Swin_T_O365.yaml. We replaced BACKBONE, LANGUAGE_BACKBONE, and DATASETS in the configuration file, and we inserted hyper-parameters for training the hypernetwork. For Flickr-30k results in Tab. 4, we add a very small fine-tuning period ($\sim 3$ epochs) to obtain them (including the baseline). For more information on the pre-training dataset, we refer the readers to the MDETR (Kamath et al., 2021) and GLIP (Li et al., 2022) papers.

In addition, the implementation of Eq. 6 and Eq. 7 may not be straightforward at first glance. We provide a PyTorch style pseudo code in this section to give an example for Eq. 6. The implementation of Eq. 7 can be done similarly.

```python
for i in range(num_layers):
    # cross_modal_shared_index_list contains indices of shared weights for all layers
    if cross_modal_shared_index_list[i].nelement() != 0:

        # shared_index contains indices of shared weights for the current layer
        shared_index = cross_modal_shared_index_list[i]

        # apply the constraint for vision pruning vectors
        # vision_pruning_vector_list contains all pruning vectors.
        vision_pruning_vector_clone = vision_pruning_vector_list[i].clone()
        # apply the constraint defined in Eq.6
        vision_pruning_vector_clone[shared_index[i]] = 1
        vision_pruning_vector_list[i] = vision_pruning_vector_clone.clone()

        # the same process is applied for text pruning vectors
        text_pruning_vector_clone = text_pruning_vector_list[i].clone()
        text_pruning_vector_clone[shared_index[i]] = 1
        text_pruning_vector_list[i] = text_pruning_vector_clone.clone()
```

[1]https://github.com/ashkamath/mdetr and https://github.com/microsoft/GLIP

## A.4 STUDY OF DIFFERENT SETTINGS

| Settings | AP | $AP_{50}$ | $AP_{75}$ | #PT | #PB |
|---|---|---|---|---|---|
| Backbones: CLIP+DeiT-Small | | | | | |
| Baseline | 17.3 | 28.7 | 17.9 | 83M | 63M |
| $\gamma = 1 \times 10^{-4}$ | 16.7 | 27.6 | 17.1 | 57M | 37M |
| $\gamma = 5 \times 10^{-4}$ | 17.2 | 28.6 | 17.6 | 57M | 37M |
| $\gamma = 1 \times 10^{-3}$ | 17.2 | 28.5 | 17.5 | 57M | 37M |
| $\lambda = 2$ | 17.0 | 28.3 | 17.3 | 57M | 37M |
| $\lambda = 6$ | 17.2 | 28.6 | 17.6 | 57M | 37M |
| $\lambda = 10$ | 16.3 | 27.4 | 16.5 | 57M | 37M |
| $|D_{\text{sub}}| = 0.01|D|$ | 16.4 | 27.4 | 16.9 | 57M | 37M |
| $|D_{\text{sub}}| = 0.10|D|$ | 17.2 | 28.6 | 17.6 | 57M | 37M |
| $|D_{\text{sub}}| = 0.20|D|$ | 17.3 | 28.7 | 17.7 | 57M | 37M |
| $|B| = 2$ | 16.8 | 28.1 | 17.2 | 57M | 37M |
| $|B| = 3$ | 17.2 | 28.6 | 17.6 | 57M | 37M |
| $|B| = 6$ | 16.4 | 27.5 | 16.7 | 57M | 37M |

Table S3: COCO object detection results. We use GLIP with CLIP+DeiT-Small as the backbone. The results are lower than Tab. 3 since the pre-training dataset is smaller due to the pre-training costs. Rows with blue color are our default settings.

To see how different choices of hyper-parameters affect our method, we adjust $\lambda$, $\gamma$, $|D_{\text{sub}}|$ (the number of samples of $D_{\text{sub}}$) and $|B|$ (the number of blocks) for our method. The overall pretraining cost is quite high for GLIP and MDETR. As a result, we only use Flickr-30k (Plummer et al., 2015) as the pre-training dataset. The experiments are conducted on GLIP with COCO object detection as the evaluation task. All other settings are the same as Tab. 3.

$\gamma$ is used to control the regularization strength of $\mathcal{R}_w$ in Obj. 11. From Tab. S3, we can see that a smaller $\gamma$ will negatively affect the final results, and this is probably because weights from different layers and different modalities are not well aligned, which creates difficulty for fine-tuning. This observation is consistent with the results of 'BWSP w/o $\mathcal{R}_w$' ($\gamma = 0$) in Tab. 1 and Tab. 2, which all indicates that poor alignment will be harmful to the final performance.

$\lambda$ is used to control the regularization strength of the parameter constraint $\mathcal{R}(P(m, s), pP_{\text{total}})$ in Obj. 9. From Tab. S3, we can see that a too large $\lambda$ ($\lambda = 10$) is harmful to our method. This is because a too large $\lambda$ will ignore the pre-training loss $\mathcal{L}_{\text{pre-training}}$ and only focus on reducing the number parameters. As a result, it hinders the hypernetwork from finding the ideal sharing and pruning structures.

$|D_{\text{sub}}|$ controls the number of samples in $|D_{\text{sub}}|$. From Tab. S3, we can see that increasing the size of $|D_{\text{sub}}|$ is beneficial to the final performance. At the same time, it increases the pre-training costs. If $|D_{\text{sub}}|$ is too small, it will definitely negatively affect the results, as shown in the table. $|D_{\text{sub}}| = 0.10|D|$ creates a good trade-off between pre-training costs and the final performance.

$|B|$ controls the number of blocks for cross-layer sharing. We tested three settings, and the results are shown in Tab. S3. When $|B| = 2$, the final model has fewer unique base weights, which restricts the weight diversity. When $|B| = 6$, the potential compression rate is limited. Clearly, when $|B| = 3$, we can have a good trade-off between the weight diversity and the potential compression rate for the text transformer.

