# OpenReview forum: "Learning to Jointly Share and Prune Weights for Grounding Based Vision and Language Models"
_ICLR.cc/2023/Conference — ICLR 2023 poster_

### Official Review · Reviewer_p7Rq · 2022-10-24

**Confidence:** 3
**Correctness:** 3
**Technical Novelty And Significance:** 3
**Empirical Novelty And Significance:** 3
**Recommendation:** 6

**Clarity, Quality, Novelty And Reproducibility:**

This paper shows  a clear description of the proposed framework and the evaluation of the method on several vision and language tasks and the framework is evaluated on several vision and language tasks.
For reproducibility, it seems hard to reproduce the method without public code.


**Strength And Weaknesses:**

Strength
- The paper is well organized and convincing
- The paper proposes a novel framework for weight sharing and pruning of transformer backbones
- The method achieves a prominent trade-off between the number of parameters and the accuracy
- The method is tested on various vision and language grounding tasks, including Referring Expression Comprehension, Visual Question Answering, and Object Detection

Weakness
- The paper does not provide a detailed analysis of the proposed method
- Also, they only share part of hyper-parameters which makes the result hard to reproduce
- Lack of justification for experimental setup, hyper-parameter selection for model and baseline.

**Summary Of The Paper:**

The paper proposed a framework to perform weight sharing and pruning across two transformer backbones and within the same transformer backbone. The framework is evaluated on vision and language tasks, including Referring Expression Comprehension (REC), Visual Question Answering (VQA), and Object Detection. The results show that the proposed framework achieves a prominent trade-off between number of parameters and accuracy.

**Summary Of The Review:**

The paper is well written and easy to follow and the proposed method is clear and well motivated.
The experiments are well designed and the results are convincing, but the paper could provide a more detailed analysis of the proposed method.

---

> ### Author Response · Authors · 2022-11-16
> **Response to Reviewer p7Rq**
>
> Thanks for your valuable comments. We list our response below.
>
> **Question 1.** The paper does not provide a detailed analysis of the proposed method.
>
> **Answer.** Thanks for your comments. We add a new section in the supplementary material to analyze different settings. We test different hyper-parameters including $\gamma$, $\lambda$, $|D_{\text{sub}}|$ (the number of samples in $D_{\text{sub}}$) and $|B|$ (the number of blocks). Due to the time limitation and the cost of pre-training, we reduce the size of the pre-training dataset when analyzing hyper-parameters extensively.
>
> **Question 2.** Also, they only share part of hyper-parameters which makes the result hard to reproduce
>
> **Answer.** We revise our paper and specify all hyper-parameters used for our method in supplementary materials, including $p$, $\lambda$, $\gamma$, and $|D_{\text{sub}}|$. We train the hypernetwork with ADAM and a constant learning rate of $0.001$ and weight decay $0.0001$. For $D_{\text{sub}}$, we randomly sample 10% of samples from $D$. For other hyper-parameters, we directly use hyper-parameters from MDETR and GLIP. For example, for the GLIP pre-training, we use the hyper-parameters given in https://github.com/microsoft/GLIP/blob/main/configs/pretrain/glip_Swin_T_O365.yaml. We replaced BACKBONE, LANGUAGE\_BACKBONE, and DATASETS in the configuration file, and we inserted hyper-parameters for training the hypernetwork. We will release our code in the final version to ensure the reproduction of our results.
>
> **Question 3.** Lack of justification for experimental setup, hyper-parameter selection for model and baseline.
>
> **Answer:** For all comparison baselines, we use the same hyper-parameter setting as our method to ensure a fair comparison between different settings.
>
> In the experimental setup, we construct three types of comparison settings. **(1).** For the first type of setting, we change the flexibility of pruning and sharing to see how it affects performance. These settings include FS, SP, LWSP, and BWSP. **(2).** For the second type of setting, we build models with different backbones so that the whole model has a similar number of parameters as the proposed BWSP. These results are used to show our method outperforms different (full) models with a similar parameter budget, which suggests our method is more parameter efficient than using smaller backbones. For example, these settings include 'DeiT-Tiny+CLIP,' 'DeiT-Small+Albert,' etc., in Tab.1 and Tab.2. **(3).** The last type of setting is `BWSP w/o ${R}_w$.' This setting is used to show the effect of ${R}_w$, and the results indicate that ${R}_w$ is crucial to the success of our method.

---

> > ### Comment · Reviewer_p7Rq · 2022-12-04
> > **Thanks**
> >
> > I appreciate the authors' efforts to make this clear.
> > The authors addressed all of my questions, and I will further discuss with other reviewer.
> > I will keep my previous rating. 6: marginally above the acceptance threshold

---

> > > ### Author Response · Authors · 2022-12-05
> > > **Thank you**
> > >
> > > Dear Reviewer p7Rq,
> > >
> > > Thank you again for your constructive and valuable comments!

---

### Official Review · Reviewer_U8mM · 2022-10-24

**Confidence:** 3
**Clarity, Quality, Novelty And Reproducibility:** The paper is well written and the pro…
**Correctness:** 3
**Technical Novelty And Significance:** 4
**Empirical Novelty And Significance:** 3
**Recommendation:** 6

**Strength And Weaknesses:**

1. Strengths:

+ The paper addresses an important topic and the proposed algorithm is well motivated. The paper is well written making it easy to follow.
+ Experiments are done carefully and the results are convincing.

2. Weaknesses:

+ Some details are missing in the current version. In page 4, the authors divide a backbone into several blocks to identify whether network weights in those different blocks can be shared. By what criteria did you divide into these blocks? What is the optimal number of blocks? There is no justification by experiments or intuition for this in the paper.

+ In page 4, after Eq. 5, the final W_q^vl is a combination of only two elements since s^l and s_{t_l} are binary. This makes the statement inaccurate.

+ In page 6, the authors mentioned that they only used a subset of training data to train HN. How does this affect the compression performance? What proportion of the subset is sufficient? Again, there are no justifications for why the authors do this.
+ The experiments conducted on the much smaller image size (224x224) compared to the original baselines, which raises doubts about the baselines' results.

+ Have the authors tried methods such as grid search to find the best hyperparameters of the backbone for that small image input?


**Summary Of The Paper:**

The paper introduces a novel compression algorithm for transformer-based vision and language models. It trains a hypernetwork to identify whether a network weight is able to share across multimodal backbones and across layers within one backbone or is able to be pruned off the original networks. Experiments show that the proposed model compression algorithm can reduce up to 40% of parameters without significant loss of accuracy in several downstream tasks.

**Summary Of The Review:**

The paper is well written and the proposed method is clearly explained.

---

> ### Author Response · Authors · 2022-11-16
> **Response to Reviewer U8mM**
>
> Thanks for your valuable comments. We list our response below.
>
> **Question 1.** Some details are missing in the current version. In page 4, the authors divide a backbone into several blocks to identify whether network weights in those different blocks can be shared. By what criteria did you divide into these blocks? What is the optimal number of blocks? There is no justification by experiments or intuition for this in the paper.
>
> **Answers.** We split blocks uniformly (each block has the same number of encoder layers), and it works well in our experiments. We would like to note that there can be a trade-off between accuracy and number of parameters w.r.t the number of encoder layers in each block. For example, if we have one block only (12 encoder layers in a block), we only have one set of base weights that can be shared with the other weights in a block. In this case, we will face an accuracy drop, which has been shown by LWSP. Another extreme case is that we have 12 blocks (1 encoder layer in a block). In this case, the compression rate of the text transformer is limited, which will negatively affect the overall results, as shown by the results of SP. Other choices of the number of blocks are $[2,3,4,6]$. If $|B|=2$ ($|B|$ is the number of blocks), the result will be close to LWSP. If $|B|=6$, the results will be close to SP. The middle number $3$ can potentially create a good trade-off between the diversity of weights in the text backbone and the compression rate. To further verify the effect of $|B|$, we add an experiment in the supplementary materials. Results in Tab.S3 indicate that we achieve an optimal trade-off when $|B|=3$.
>
> **Question 2.** In page 4, after Eq. 5, the final $W_q^{v_l}$ is a combination of only two elements since $s^l$ and $s_{t_l}$ are binary. This makes the statement inaccurate.
>
> **Answer:** Thanks for raising this question. Indeed $s^l$ and $s^{t_l}$ are binary, but they are different for different weight vectors. For example, let $i,j,k$ be three indices representing the corresponding weight vectors. In this case, if $s_{i}^l=0$ and $s_{i}^{t_l}=0$, then $W_{q[i,:]}^{v_l} = W_{q[i,:]}^{t_b}$. If $s_{j}^l=0$ and $s_{j}^{t_l}=1$, then $W_{q[j,:]}^{v_l} = W_{q[j,:]}^{t_l}$. If $s_{k}^l=1$, then $W_{q[k,:]}^{v_l} = W_{q[k,:]}^{v_l}$. As a result, the final $W_q^{v_l}$ can have weight vectors from $W_q^{t_b}$, $W_q^{t_l}$ and $W_q^{v_l}$.
>
> **Question 3.** In page 6, the authors mentioned that they only used a subset of training data to train HN. How does this affect the compression performance? What proportion of the subset is sufficient? Again, there are no justifications for why the authors do this.
>
> **Answer.** Thanks for pointing this out. We set $|D_{\text{sub}}|=0.10|D|$. We revise the paper and add the comparison of different choices of $|D_{\text{sub}}|$ in Tab.S3. Basically, a small size of $|D_{\text{sub}}|$ ($0.01|D|$) is harmful to the final results (near $1.0$ lost in AP on COCO object detection). On the other hand, increasing the size of $|D_{\text{sub}}|$ has positive impacts, but the difference is not large. Overall speaking, the pre-training process is time-consuming, so we do not want to use a large $|D_{\text{sub}}|$, and $|D_{\text{sub}}|=0.10|D|$ provides a good trade-off between the performance and pre-training costs.
>
>
> **Question 4.** The experiments conducted on the much smaller image size (224x224) compared to the original baselines, which raises doubts about the baselines' results.
>
> **Answers.** In addition to $224\times224$ pixels images, we also conducted experiments with 384x384 pixels images using the 226M parameters model. With this model without any sharing or pruning (baseline), we get 85.9\% accuracy on RefCOCO val split. With weight sharing and pruning (ours), we achieve 86.1\% accuracy on RefCOCO val split. We note that with our weight sharing and pruning method, we reduce the number of parameters of the baseline model to 136M parameters. These results are within $\sim$1\% range of the results presented by MDETR paper on $800\times1333$ pixels images. In our final paper, we will include results using $384\times384$ pixels images in RefCOCO+ and RefCOCOg datasets.
>
> **Question 5.** Have the authors tried methods such as grid search to find the best hyperparameters of the backbone for that small image input?
>
> **Answer.** We did not search for hyper-parameters when training our models at resolution $224\times224$. We simply follow training hyper-parameters from MDETR and GLIP. To the best of our knowledge, changing the resolution may not have a large impact on hyper-parameters, as shown in [1]. In addition, using the original hyper-parameters will be beneficial for reproduction. We will also open-source our code in the final version.
>
> [1] Hugo Touvron, Andrea Vedaldi, Matthijs Douze, and Herv ́e J ́egou. Fixing the train-test resolution discrepancy. Advances in neural information processing systems, 32, 2019.

---

### Official Review · Reviewer_FNmr · 2022-10-25

**Confidence:** 3
**Correctness:** 4
**Technical Novelty And Significance:** 4
**Empirical Novelty And Significance:** 4
**Recommendation:** 8

**Clarity, Quality, Novelty And Reproducibility:**

Although I am not the expert in model pruning, I think this paper is novel and should attract multiple readers in model pruning and vision+language communities.
As I didn't have experience in implementing the model pruning approach, I cannot judge the reproducibility of this paper.

**Strength And Weaknesses:**

The paper is interesting to read and the proposed idea is novel.
I just have several quick questions and comments.
1. The proposed approach seems could be apply not only on the image-text retrieval model, but also on single modal models (e.g. image only model and text only model). I wonder why would the author choose the image-text retrieval model?
2. Just curious, as the ConvNet models also shares unified structure across the model, would the proposed approach work on the ConvNet models?
3. How to set the base layers? Is it set heuristically?
4. I might miss some details, but what is the 'z' in Eq.8 in implementation?
5. Just curious, how to enforce the Eq.6 and Eq.7 during training?
6. I wonder did the author observe any training instabilities for image-text retrieval model pre-training?

**Summary Of The Paper:**

The paper studies the weight pruning and weight sharing for image-text retrieval model. Contrary to previous works that applies the pruning and sharing on the layer level, the main idea of this paper is to apply weight sharing and weight pruning on the weight level. Therefore, the pruned model could achieve better compression rate and performance. The pruning and sharing is controlled by a hyper network, which is trained jointly with the image-text retrieval model. The proposed approach achieves better performance and better compression rate than the previous state-of-the=art.

**Summary Of The Review:**

Given the improved performance and the novelty, I would recommend the rating of 8.

---

> ### Author Response · Authors · 2022-11-16
> **Response to Reviewer FNmr**
>
> Thanks for your valuable comments. We list our response below.
>
> **Question 1.** The proposed approach seems could be apply not only on the image-text retrieval model, but also on single modal models (e.g. image only model and text only model). I wonder why would the author choose the image-text retrieval model?
>
> **Answers.** The cross-modal sharing in our method builds on similar structures of text and image transformers, and it plays an important role in our method. If we apply our method to text or image only models, we can hardly achieve a high compression rate without cross-modal sharing. In addition, compared to image/text benchmark tasks, the tasks from image-text grounding models have more potential practical impacts. So we think it will be interesting to incorporate our method with image-text grounding models.
>
> **Question 2.**  Just curious, as the ConvNet models also shares unified structure across the model, would the proposed approach work on the ConvNet models?
>
> **Answers.** It is possible to apply our method for cross-modal ConvNet models as long as they have similar architectures, and an example task could be action recognition (RGB + Optical flow). The overall range of ConvNets tasks might be smaller than transformers since it's hard to use architecturally similar ConvNets for image-text tasks.
> %We will explore how to use our method on ConvNets and related tasks in our future works.
>
> **Question 3.**  How to set the base layers? Is it set heuristically?
>
> **Answers.** Currently, we heuristically set the base layers as the first layer within each block. There are other potential choices of base layers or even learning to set base layers, and we will leave this part for future work.
>
> **Question 4.** I might miss some details, but what is the 'z' in Eq.8 in implementation?
>
> **Answers.** z is the input to the hypernetwork, and it is used to provide inputs to GRU in the hypernetwork, and more details can be found in Tab.S1.
>
> **Question 5.** Just curious, how to enforce the Eq.6 and Eq.7 during training?
>
> **Answers.** The implementation is not very complicated, except that we need to use ‘clone’ to avoid in-place modification errors during gradient calculations. We add a PyTorch like pseudo code for the cross-modal sharing (Eq.6) as an example in supplementary materials.
>
> **Question 6.** I wonder did the author observe any training instabilities for image-text retrieval model pre-training?
>
> **Answers.** We did not observe instabilities in the pre-training process. We hypothesize that this is because we progressively align model weights in the training process instead of directly changing its architecture.

---

### Author Response · Authors · 2022-11-16
**Summary of the Revision**

We want to thank all reviewers for their time and efforts in reviewing our paper. We revise our paper according to comments from all reviewers. In the supplementary material, we add an analysis of different choices of hyper-parameters for our method in the section **Study of Different Settings**. We also revise the section of **Additional Details for Experiments** in the supplementary material to include more implementation details.

---

### Decision · Program_Chairs · 2023-01-20

**Decision:**

Accept: poster

**Justification For Why Not Higher Score:**

Although the proposed idea seems novel and the work is well executed, most techniques were borrowed from exiting literature and the technical depth is not particularly strong.

**Justification For Why Not Lower Score:**

N/A

**Metareview: Summary, Strengths And Weaknesses:**

The paper proposed a novel compression algorithm for transformer-based vision and language models and empirically show that it can reduce up to 40% of parameters without significant loss of accuracy in several downstream tasks. Reviewers generally agree the work is interesting, the proposed idea is novel and well motivated, the empirical results are convincing. There are some minor questions about the experiments and ablation studies and clarity of the presentation, authors made proper responses to address them during rebuttal. Overall, this is an interesting and practically useful paper that is recommended to accept.



**Note From Pc:**

if the above contains the word "oral" or "spotlight" please see: "oral" presentation means -> notable-top-5% and "spotlight" means -> notable-top-25%. As stated in our emails, we are disassociating presentation type from AC recommendations